# Lymphangiogenesis Guidance Mechanisms and Therapeutic Implications in Pathological States of the Cornea

**DOI:** 10.3390/cells12020319

**Published:** 2023-01-14

**Authors:** Mehul Patnam, Sunil R. Dommaraju, Faisal Masood, Paula Herbst, Jin-Hong Chang, Wen-Yang Hu, Mark I. Rosenblatt, Dimitri T. Azar

**Affiliations:** 1Department of Ophthalmology and Visual Sciences, Illinois Eye and Ear Infirmary, College of Medicine, University of Illinois at Chicago, Chicago, IL 60612, USA; 2Department of Urology, University of Illinois at Chicago, Chicago, IL 60612, USA

**Keywords:** cornea, lymphangiogenesis, lymphatic vessel, lymphatic endothelial cell, guidance, inflammation, (lymph)angiogenic privilege, vascular endothelial growth factor (VEGF)

## Abstract

Corneal lymphangiogenesis is one component of the neovascularization observed in several inflammatory pathologies of the cornea including dry eye disease and corneal graft rejection. Following injury, corneal (lymph)angiogenic privilege is impaired, allowing ingrowth of blood and lymphatic vessels into the previously avascular cornea. While the mechanisms underlying pathological corneal hemangiogenesis have been well described, knowledge of the lymphangiogenesis guidance mechanisms in the cornea is relatively scarce. Various signaling pathways are involved in lymphangiogenesis guidance in general, each influencing one or multiple stages of lymphatic vessel development. Most endogenous factors that guide corneal lymphatic vessel growth or regression act via the vascular endothelial growth factor C signaling pathway, a central regulator of lymphangiogenesis. Several exogenous factors have recently been repurposed and shown to regulate corneal lymphangiogenesis, uncovering unique signaling pathways not previously known to influence lymphatic vessel guidance. A strong understanding of the relevant lymphangiogenesis guidance mechanisms can facilitate the development of targeted anti-lymphangiogenic therapeutics for corneal pathologies. In this review, we examine the current knowledge of lymphatic guidance cues, their regulation of inflammatory states in the cornea, and recently discovered anti-lymphangiogenic therapeutic modalities.

## 1. Introduction

The cornea sits on the anterior-most surface of the eye and serves to refract light, transmit it to the retina, and act as a physical barrier between the eye and its external environment. From anterior to posterior, the cornea consists of five layers: the epithelium, Bowman’s membrane, stroma, Descemet’s membrane, and endothelium [1]. Corneal transparency, a property which is crucial for the cornea’s refractive and transmissive capacities, is primarily maintained by precise stromal organization and the absence of blood and lymphatic vasculature [2,3].

Normally, corneal avascularity is maintained by an active process that favors the production of anti-(lymph)angiogenic factors over pro-(lymph)angiogenic factors [4]. The absence of blood and lymphatic vessels that is maintained via this balance is termed “(lymph)angiogenic privilege”. Through extensive research, the mechanisms of corneal hemangiogenesis have been well described, but in comparison, the characterization of corneal lymphangiogenesis lags far behind. In normal states, lymphatic vessels extend to the limbus, a transition zone between the cornea and sclera, and form a circular network around the cornea without invading the corneal tissue [5]. Corneal avascularity is already established at birth via the anti-(lymph)angiogenic functions of the limbus and collagen fibers of the corneal stroma [6,7]. Despite the establishment of (lymph)angiogenic privilege during embryonic development, the adult cornea possesses the ability to undergo revascularization in response to inflammatory insult, establishing it as a tissue with lymphatic plasticity [8]. Lymphatic revascularization is regulated by various molecular players that influence the spatiotemporal fate of lymphatic endothelial cells (LECs). This directed cell migration and lymphatic endothelial guidance allows for the patterning of lymphatic vasculature.

Inflammation-induced damage can disrupt strict regulatory processes, resulting in the pathological invasion of blood and lymphatic vessels into the avascular cornea [9]. Lymphangiogenesis describes the expansion of the lymphatic network, including the development of new lymphatic vessels from preexisting ones and de novo lymphatic vessel formation. It is triggered when LECs respond to growth factors or biomechanical stimuli [10]. Corneal lymphangiogenesis in particular has been implicated in a variety of ocular pathologies, such as corneal graft rejection, dry eye disease, infectious keratitis, alkali burn, and ocular tumors [11]. In dry eye disease, corneal lymphangiogenesis mediates the trafficking of foreign antigens to regional lymph nodes and initiates a deleterious immune response [12]. The presence of preexisting lymphatic vessels in the cornea is also associated with a decreased graft survival rate following corneal transplantation [13]. Despite this evidence of pathological corneal lymphangiogenesis, recent studies have identified novel beneficial roles of lymphangiogenesis in the resolution of corneal disease. Administration of antibodies to block corneal lymphangiogenesis in a corneal incision injury mouse model results in sustained increases in central corneal thickness, highlighting the role of lymphangiogenesis in the resolution of corneal edema [14]. In the late stages of a murine bacterial keratitis model, corneal lymphangiogenesis ameliorates corneal edema and prevents further decline in the clinical keratitis score [15]. As corneal lymphangiogenesis has been observed to both exacerbate pathology and accelerate healing in a disease-specific manner, tailored approaches to modulating lymphangiogenesis may reveal therapeutic benefits.

Although corneal avascularity describes a lack of both blood vessels and lymph vessels, the medical literature has predominantly focused on corneal hemangiogenesis and anti-angiogenic therapies. To date, more than 25 anti-angiogenesis drugs that target a variety of different signaling pathways have been approved by the Food and Drug Administration (FDA) [16]. While many of these drugs have now been shown to additionally influence lymphangiogenesis in preclinical models, emphasis on regulators of lymphangiogenesis in clinical trials is scarce [17]. As such, there is much room for investigation into anti-lymphangiogenic therapeutic strategies.

This review summarizes the known lymphangiogenic guidance cues and the mechanisms by which they influence the directional migration of LECs and lymphatic network patterning in the cornea. A focused understanding of lymphatic endothelial guidance in pathological states of the cornea can highlight potential targets for anti-lymphangiogenic therapeutics.

## 2. Basic Principles of Lymphatic Network Development & Patterning

### 2.1. Lymphatic Network Origin and Development

Since the advent of lymphatic-specific markers for microscopic visualization of lymph vessels, several endogenous and exogenous factors have been identified that impact lymphangiogenesis in varying ways. Lymphangiogenesis is a complex process involving lymphatic alignment, LEC proliferation, lymphatic vessel sprouting, and lymphatic vessel deflection (Figure 1). Lymphatic guidance cues are signals that direct LEC migration or vessel formation via attractive or repulsive signals. Herein, we differentiate between direct lymphatic endothelial guidance mechanisms and vascular endothelial growth factor (VEGF)-mediated indirect guidance mechanisms.

While most lymphatic networks are generated during embryonic development, they can undergo drastic changes in adults, growing in length and caliber [19,20]. A wide variety of growth factors and cytokines are known to influence lymphangiogenesis. The primary growth factors promoting lymphangiogenesis are VEGF-C and VEGF-D [21]. Stimulation of vascular endothelial growth factor receptor-2 and -3 (VEGFR-2 and VEGFR-3), the primary receptors of VEGF-C and VEGF-D, promotes LEC proliferation, migration, and stabilization [18]. VEGF-A, the primary growth factor promoting hemangiogenesis, has been indirectly implicated in lymphangiogenesis by promoting the recruitment of VEGFR1-positive, cluster of differentiation 11b^+^ (CD11b^+^) macrophages that secrete VEGF-C and -D [22,23,24]. Other crucial pro-lymphangiogenic factors include fibroblast growth factor (FGF), platelet-derived growth factor (PDGF) [25], angiopoietin [26], and several others.

### 2.2. Tip and Stalk Cell Theory of Lymphangiogenesis

The process of lymphangiogenesis requires the complex directional coordination of a variety of cells, similar to the better studied process of hemangiogenesis. Currently, the most well-described growth mode for lymphangiogenesis in non-developmental settings is LEC sprouting from pre-existing vessels [24,27,28]. One hypothesis is that these sprouts grow and proliferate to form new lymphatic vessels, guided by migrating tip cells [29,30]. Tip cells display fine filopodia and cellular protrusions that search the pericellular environment for guidance cues [31]. These molecular cues act as either attractive or repulsive signals that additively influence the directional growth of the budding lymphatic vessel [32,33,34,35]. In vascular hemangiogenesis, a VEGF-A gradient guides vascular endothelial tip cells [36,37]. Additionally, while tip cells are important for guiding the direction of blood vessel growth, the majority of vascular endothelial cell proliferation in the sprouting blood vessel occurs in stalk cells, which lag just behind the tip cells [38].

Although the tip cell and stalk cell phenotypes in hemangiogenesis have been characterized in detail, the model has yet to be fully established in lymphangiogenesis [39]. Some evidence suggests that LECs may acquire specific tip cell phenotypes and guide growing lymphatic branches during lymphangiogenesis [18,40]. However, recent studies provided evidence against the application of this model to lymphangiogenesis. Interestingly, VEGF-C-induced fine filopodia are apparent at multiple points along the proliferating lymphatic vessel rather than being localized to tip cells as in hemangiogenesis [35]. In a corneal lymphangiogenesis mouse model, the proportion of proliferating cells observed at the sprout extremities was double that at the sprout stalk [41]. Though the “tip” and “stalk” cell theory of hemangiogenesis has been implicated in lymphangiogenesis, these data indicate that a complete translation of the model is not accurate and that different guidance mechanisms underlie hemangiogenesis and lymphangiogenesis.

Thus, to identify therapeutic targets for modulation of corneal lymphangiogenesis, it is critical to understand not only the growth factors that mediate lymphatic vessel proliferation and inhibition but also the specific guidance mechanisms governing lymphatic network patterning in the cornea.

## 3. Corneal Lymphangiogenesis Guidance Mechanisms

In this section, we first present the background information and in vitro findings regarding regulators of lymphatic vessel development. We follow this with an analysis of corneal lymphangiogenesis guidance in pathological states. Factors contributing to lymphangiogenesis guidance and their effects on LECs are summarized in Table 1. The progression of lymphangiogenesis, including the significant guidance molecules that influence LEC fate at each step, is depicted in Figure 2.

**Table 1 cells-12-00319-t001:** Guidance molecules that mediate lymphangiogenesis.

Molecular Family	Molecule	Target or Ligand	Effect on Lymphatic Endothelial Guidance	References
VEGF	VEGF-A	VEGFR-1,2 Notch signaling	Induces LEC proliferation. Recruits VEGF-C/D-producing macrophages to sites of injury. Internalizes VEGF-C/VEGFR-2 complexes in vascular endothelial cells.	[18,23,36,37,38,42,43,44,45,46,47,48]
	VEGF-C	VEGFR-2,3	Expands the lymphatic vascular network via LEC migration, proliferation, and sprouting. Essential for lymphatic network formation from embryonic cardinal vein. Influences directional migration of lymphangioblasts. Primary driver of lymphangiogenesis.	
	VEGF-D	VEGFR-3	Expands the lymphatic vascular network via LEC migration, proliferation, and sprouting.	
FGF	bFGF	FGFR	Increases the secretion of VEGF-C from vascular endothelial cells. Directly binds to LECs and promotes LEC migration, proliferation, and survival. Pro-lymphangiogenic effects inhibited upon interaction with LYVE-1.	[49,50,51]
Neuropilin	Neuropilin-2	VEGF-C	Acts as a coreceptor for VEGFR-3, promoting LEC migration and sprouting. Encourages lymphatic endothelial tip cell extension and prevents tip cell retraction during sprouting. Highly concentrated in cells at the leading tip of growing lymphatic sprouts.	[52,53,54,55]
Ephrin	EphrinB2	EphB4	Promotes maturation of lymphatic valves both during valvulogenesis and post-injury. Promotes VEGF-induced LEC migration and lymphatic tube formation.	[56,57,58,59,60]
Wnt proteins	Wnt5a	FZD3, RYK, β-catenin	Promotes maturation of lymphatic valves both during valvulogenesis and post-injury. Elongates lymphatic networks.	[61,62,63]
Netrin	Netrin-4	Unc5 Neogenin	Promotes LEC migration, proliferation, adhesion, and tube formation.	[64,65]
Slit	Slit2	Robo1	Stimulates LEC migration and tube formation.	[66,67]
		Robo4	Induces VEGFR-3 internalization in LECs. Inhibits the activation of LECs by VEGF-C.	
CXCL	CXCL12	CXCR4, VEGF-C	Induces LEC migration and tube formation. Directs early trunk lymphatic network assembly.	[68,69,70]
Sphingolipids	S1P	S1PR1	Influences inward LEC migration in response to wall stress and directional LEC migration in response to fluid flow stimulus. When absent, induces VEGF-C expression.	[71,72,73,74,75,76]
Glycosaminoglycans	Hyaluronan (LMW, HMW)	LYVE-1, LYVE-2, S1P-3, ERK-1/2	Promotes lymphatic vessel sprouting and proliferation in both healthy and pathological states. Organizes lymphatic endothelium into vessel-like cell sheets, promoting lymphatic tube formation. Synergistically increases lymphatic tube formation and sprouting when administered with VEGF-C.	[8,77,78]
Integrins	α9β1	VEGF-A/C/D, fibronectin EDA, emilin-1, polydom	Promotes LEC migration, vessel sprouting, and both developmental and pathological valvulogenesis.	[79,80,81,82,83]
	α5β1		Promotes LEC sprouting and VEGF-C-mediated guidance.	
	α6β1		Promotes LEC adhesion and migration to netrin-4.	
BMP	BMP4	ALK	Inhibits guidance and neovascularization by decreasing VEGF-C/VEGFR-3 signaling.	[84,85]
	BMP9		Directs early lymphatic endothelial tip cell expansion. Low concentration enhances LYVE-1-positive LECs; high concentration enhances LYVE-1-negative LECs. Activates VEGF-A at high concentrations.	
Angiopoietin	Ang-1	Tie-1 Tie-2	Ang-2 guides sprouting of lymphatics around blood vessels. Ang-2 sensitizes LECs to inflammatory stimuli post-injury. Tie-1 facilitates early stages of developmental LEC proliferation and LEC survival.	[86,87,88]
	Ang-2			
TGFBIp		Integrins	Promotes LEC sprouting, migration, adhesion, and tube formation. Synergistically enhances stimulatory effect of VEGF-C.	[89]
Semaphorins	Sema3A	NRP1, plexinA1	Contributes to lymphatic vessel and valve morphology during development.	[90,91,92,93,94,95,96,97,98,99,100,101]
	Sema3F	NRP2, plexinA3, plexinA1	Globally suppresses LEC proliferation and sprouting in low concentrations. Overexpression causes a chemorepulsive effect on LECs.	
	Sema3G	NRP2, plexinD1, plexinA2	Locally suppresses LEC proliferation and sprouting in high concentrations. Repels LECs away from arteries and induces lymphatic vascular branching.	
	Sema7A	β1-integrin receptor	Promotes lymphatic vessel invasion, including LEC tube formation.	
Delta-like ligands	Dll4	Notch	Suppresses LEC migration and lymphatic vessel sprouting. Suppresses lymphangiogenesis via effects on VEGF-A and VEGF-C signaling. Suppresses Prox1+ LECs during embryonic development.	[32,44]

Abbreviations: FGF, fibroblast growth factor; bFGF, basic fibroblast growth factor; FGFR, fibroblast growth factor receptor; LMW, low molecular weight; HMW, high molecular weight; ERK, extracellular signal-regulated kinase; FZD3, frizzled class receptor 3; RYK, related to receptor tyrosine kinase; S1P, Sphingosine 1-phosphate; S1PR1, Sphingosine 1-phosphate receptor 1; LYVE-1, lymphatic vessel endothelial hyaluronan receptor 1; EDA, extra domain A; BMP, bone morphogenetic protein; ALK, activin receptor-like kinase; Ang, angiopoietin; Tie, angiopoietin receptor; TGFBIp, transforming growth factor-beta-induced protein; Sema, semaphorin; NRP1, Neuropilin-1; Dll, Delta-like ligand.

**Figure 2 cells-12-00319-f002:**
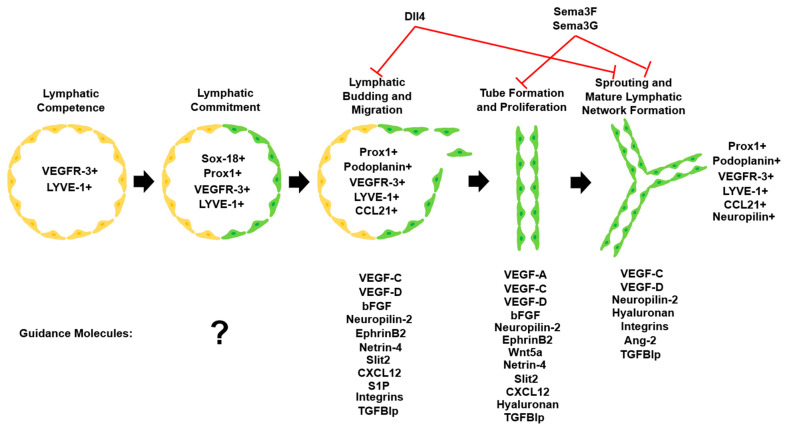
Lymphatic vascular system commitment and development. Upon commitment to the lymphatic lineage, LECs undergo migration, proliferation, and sprouting to form the mature lymphatic network. Important lymphatic markers that are detectable at each step and various guidance molecules influence LEC fate during lymphangiogenesis [102]. Adapted with permission.

### 3.1. VEGF Family

The VEGF family and its associated cognate receptors (VEGFRs) are among the most important regulators of lymphangiogenesis. VEGFRs are primarily expressed in endothelial cells, and VEGFs are primarily secreted by inflammatory cells such as macrophages [103]. VEGF-A, -C, and -D in particular have been identified as key factors involved in inducing and sustaining hemangiogenesis and lymphangiogenesis [11]. VEGFR-2 and -3 are key endothelial cell receptors that respond to VEGF-A, -C, and -D [18]. VEGF-A interacts with both VEGFR-1 and -2, VEGF-C with both VEGFR-2 and -3, and VEGF-D with both VEGFR-2 and -3 (Figure 3) [18].

Cellular responses to VEGF-C/-D binding of VEGFR-3 are the most important regulators of lymphangiogenesis [105]. During embryogenesis, VEGFR-3 is expressed on all blood vessels but becomes largely restricted to LECs in adulthood [106]. VEGFR-3 expression is downregulated as blood vessels commit to the LEC lineage [105]. In embryogenesis, VEGF-C is essential for the sprouting and formation of lymphatic networks from the embryonic cardinal vein [107]. Recently, Wang et al. described the importance of VEGF-C activation in the directional migration of parachordal lymphangioblasts in zebrafish, highlighting the role of VEGF-C/VEGFR-3 signaling in guiding lymphangiogenesis [48].

The VEGF-A/VEGFR-2 interaction is known as the primary driver of hemangiogenesis across various embryonic and post-natal contexts. VEGF-A/VEGFR-2 interactions also promote lymphangiogenesis, albeit to a lesser extent than VEGF-C/-D binding to VEGFR-3. Despite its positive effect on lymphatic vessel proliferation, VEGFR-2 activation does not impact sprouting of new lymphatic vessels [42]. VEGF-A can promote lymphangiogenesis indirectly by binding VEGFR-2 on macrophages and inducing recruitment of VEGF-C/-D-producing macrophages to sites of injury [23,108]. VEGF-A may also regulate lymphangiogenesis by inducing the internalization of VEGF-C/VEGFR-2 complexes in vascular endothelial cells [109].

In corneal lymphangiogenesis induced by pathological stimuli, VEGF-C/VEGFR-3 signaling has been shown to be the primary driver of lymphangiogenesis, with VEGF-A/VEGFR-2 signaling playing a secondary role [45,46,47,110]. In contrast, in an in vivo model of corneal herpes simplex virus 1 (HSV-1) infection, lymphangiogenesis appeared to be solely driven by VEGF-A/VEGFR-2 signaling in a macrophage-independent manner, with VEGF-A being secreted by infected corneal epithelial cells [111]. In the same study, VEGFR-3 inhibition significantly decreased corneal lymphangiogenesis in a penetrating suture model, suggesting that corneal lymphangiogenic pathways are context-dependent. In the cornea, VEGF-C is primarily expressed by macrophages as well as corneal epithelial cells [112]. It is thought that during corneal inflammation, macrophages are recruited to the area of injury via cytokines where they can induce hemangiogenesis and lymphangiogenesis as needed [11]. VEGF-A is also capable of recruiting VEGF-C- and VEGF-D-secreting macrophages to promote corneal lymphangiogenesis [113]. In summary, VEGF-A/VEGFR-2 and VEGF-C/VEGFR-3 signaling play important roles in initiating and facilitating LEC sprouting, LEC proliferation, and expansion of the corneal lymphatic network. However, the neovascularization process in the cornea has certain unique aspects, such as context-dependent drivers of lymphangiogenesis and physiologic lymphangiogenic privilege, which must be considered. Several other endogenous factors influence LEC fate and lymphatic vessel patterning in the cornea indirectly through modulation of VEGF/VEGFR signaling.

### 3.2. bFGF

Basic fibroblast growth factor (bFGF or FGF-2) has been shown to indirectly impact lymphangiogenesis guidance primarily by increasing VEGF-C and -D expression [4,51]. However, a potential direct mechanism for bFGF-mediated lymphangiogenesis has also been suggested. bFGF can bind to LECs and promote LEC migration, proliferation, and survival even in the presence of an anti-VEGFR-3 blocking antibody, suggesting that bFGF possesses prolymphangiogenic properties independent of VEGF-C/VEGFR-3 signaling [49]. A combined effect of bFGF and VEGF-C on lymphangiogenesis was observed in a tumor microenvironment, with stimulation by both factors inducing greater lymphangiogenesis than either factor individually [50]. In the same study, VEGFR-3-mediated formation of tip cells was found to be critical to the induction of lymphangiogenesis by bFGF. The lymphatic endothelial-specific marker lymphatic vessel endothelial hyaluronan receptor 1 (LYVE-1) interacts with bFGF to inhibit its stimulatory effects on lymphatic vessel migration, proliferation, and invasion in the cornea [114].

Hajrasouliha et al. described distinct patterns of lymphatic vessel growth in a murine model of corneal lymphangiogenesis. Upon implantation of a bFGF pellet in the cornea, blood vessels grew towards the pellet from the limbal region of the cornea closest to the pellet [115]. Interestingly, despite proximity to one side of the limbal periphery, lymphatic vessels primarily grew towards the pellet from the limbal region of the cornea opposite to the implantation site. In line with these findings, VEGF-A and -C expression was increased the most in the regions closest to the pellet where blood vessels grew while VEGF-C and -D were upregulated most in the areas opposite to the pellet where lymphatic vessels grew. The differential regional expression of VEGF isoforms in response to bFGF-induced lymphangiogenesis suggests that localized lymphangiogenesis guidance depends on the specific site of corneal inflammation. This warrants further investigation into the regional variance of guidance mechanisms in specific disease states.

### 3.3. Neuropilin-2

Neuropilin-2 (NRP2), originally identified as an axon guidance molecule, is now also implicated in the formation of small lymphatic vessels and capillaries; it acts as a co-receptor for VEGFR-3 [52,116]. The high-affinity monoclonal antibody anti-NRP2 blocks NRP2 from binding to VEGF-C and inhibits VEGF-C-related LEC migration and guidance signaling pathways [53]. It was also found that blocking VEGF-C binding to NRP2 inhibits sprouting of new lymphatic endothelial tip cells, which ultimately reduces vessel sprouting and decreases vessel branching; therefore, NRP2 encourages lymphatic endothelial tip cell extension and prevents tip cells from stalling or retracting during the sprouting process [54]. This is consistent with the fact that NRP2 is highly concentrated in the cells at the leading tip of growing lymphatic sprouts [53].

In the context of corneal inflammation, inhibition of NRP2 improves survival of corneal allografts due to an inhibited immune response without an effect on macrophage recruitment [55]. Current studies on therapies targeting NRPs to selectively curtail corneal lymphangiogenesis by targeting VEGFR-3 expression are promising, with preliminary results showing that NRP2 knockdown prevents increased corneal lymphatic vessel growth following lipopolysaccharide (LPS)-induced corneal inflammation [117]. Additionally, NRP2 knockdown prevents VEGFR-3 upregulation and inhibits lymphatic vessel proliferation without impacting blood vessels, highlighting its specialized role in influencing lymphangiogenesis [117].

### 3.4. Ephrin–Eph Signaling

Ephrin–Eph signaling comprises the largest family of receptor tyrosine kinases and is critical in regulating neuron position, tissue morphology, boundary formation, cell segregation, and axon guidance [118]. Studies have demonstrated the necessity of Ephrin–Eph signaling in axon guidance for the establishment of correct synapses and left–right asymmetry [56,57]. Bidirectional signaling is an important hallmark of the Ephrin–Eph signaling system. This forward and reverse signaling has recently been implicated in lymphangiogenesis as well [119,120,121]. One study demonstrated that selective inhibition of EphB4-dependent forward signaling results in defective lymphatic valve development [58]. Blockage of membrane-anchored ephrin-B2 suppresses endothelial cell migration and tube formation in vitro in response to VEGF, and systemic administration of ephrin B2-blocking antibodies causes drastic reductions in blood and lymphatic vessel growth in xenografted mice [59].

In the cornea, ephrin-B2–EphB4 signaling has been shown to modulate the morphologic maturation of lymphatic valves during corneal lymphangiogenesis post-injury [122]. Following alkali burn, fluctuations in ephrin-B2 and EphB4 levels across various stages of lymphangiogenesis correspond with the formation of new valves. Subconjunctival injections of recombinant EphB4/Fc, a trap for the ephrin-B2 ligand, were found to disrupt the funnel-shaped valve morphology and reduce the length of valve invagination [122]. Considering the critical role of Ephrin–Eph signaling in axon guidance and the identification of novel lymphangiogenic functions, further investigation into how Ephrin–Eph signaling guides lymphangiogenesis in the cornea is merited.

### 3.5. Wnt/β-Catenin

The canonical Wnt pathway is known to be involved in lymphatic valve formation as well as lymphatic vessel patterning with β-catenin. Shear stress, a known inducer of lymphatic vessel maturation, triggers the Wnt/β-catenin pathway in LECs [63]. Subsequent activation of Forkhead box protein C2 (FOXC2), a lymphedema-associated transcription factor, downstream of this pathway promotes lymphatic vessel patterning in mice [63]. Apart from this function, the Wnt system of proteins has been implicated in other important physiological functions, such as cell fate determination and stem cell maintenance, with indirect participation in guidance mechanisms [123]. Wnt5a is also responsible for guiding the appropriate development of initial lymphatics and lymphatic collectors by modulating lymphatic morphogenesis. The absence of Wnt5a results in ruffled LECs with abnormal protrusions as well as dilated lymphatics that abnormally contain blood cells [62].

Under corneal inflammatory conditions, knockout of Wntless, a gene that encodes a Wnt receptor, in mice results in reduced lymphatic area and ingrowth from the limbal arcades [61]. Interestingly, inhibition of lymphangiogenesis via Wnt5a knockout was found to be coupled with decreased VEGF-C production by macrophages, suggesting that Wnt indirectly regulates corneal inflammatory lymphangiogenesis via VEGF-C/VEGFR-3 signaling [61]. Beyond this, the involvement of Wnt proteins in the guidance of corneal lymphangiogenesis remains underexplored.

### 3.6. Netrins

Netrins are a family of proteins known to guide axons during neural development via binding to their canonical receptors from the Unc5, Deleted Colon Cancer (DCC), and neogenin families [65]. A proangiogenic role for the netrin-1 receptor Unc5B has recently been discovered. Deletion of the gene encoding Unc5B in the mouse embryonic endothelium disrupts formation of the placental arteriolar network [124]. Interestingly, knockdown of netrin-1a in zebrafish prevents formation of the parachordal vessel, a blood vessel that critically guides LEC migration [124,125].

Though knowledge of the role of Netrin-1/Unc5B signaling in lymphangiogenesis is sparse, a different member of the netrin family has recently been shown to influence lymphangiogenesis. Netrin-4 acts as a chemoattractant and mitogen in lymphangiogenesis in vitro, inducing LEC migration, proliferation, adhesion, and tube formation [64]. Inactivation of netrin-1 receptors Unc5B and neogenin was unable to abolish the pro-lymphangiogenic response, suggesting that netrin-4 impacts LEC fate via an alternative guidance mechanism [64]. Instead, netrin-4-mediated phosphorylation of various intracellular signaling pathways, primarily the phosphoinositide 3-kinase (PI3K) and Akt signaling pathways, is crucial to its pro-lymphangiogenic properties. Additionally, the α6β1 integrin binds netrin-4 and is expressed in lymphatic endothelium, highlighting a potential therapeutic target for inhibiting netrin-induced lymphangiogenesis [83]. In vivo, netrin-4 overexpression in mouse skin increases lymphatic vessel density without prior macrophage recruitment, suggesting that promotion of lymphangiogenesis via netrin-4 likely occurs via a macrophage-independent mechanism [64].

The role of netrin-4 in corneal lymphangiogenesis has yet to be fully determined. Interestingly, netrin-4 expression was not observed in corneal lymphatic vessels, and accordingly, netrin-4 deletion did not impact lymphatic vessel area or VEGF-C expression in mouse corneas post-injury [126]. This in vivo result contradicts the pro-lymphangiogenic functions of netrin-4 in vitro and underscores the need for further investigation into netrin-4’s role in guiding LECs.

### 3.7. Slit/Robo

The Slit proteins are a family of guidance cues that bind Roundabout (Robo) receptors to modulate neuronal, leukocytic, and endothelial migration. One study found that Slit2–Robo1 interactions stimulate migration and tube formation in LECs and that transgenic overexpression of Slit2 in a mouse tumor model increases tumor lymphangiogenesis and metastasis [66]. Yu et al. observed that in vitro, the active fragment of Slit2 (Slit2N) inhibits LEC migration and tube formation [67]. Furthermore, Slit2–Robo4 activation induces VEGFR-3 internalization in cultured LECs, inhibiting the activation of LECs by VEGF-C [67].

While Slit/Robo signaling has not been directly studied in corneal lymphangiogenesis, findings from corneal hemangiogenesis suggest an indirect impact of Slit/Robo signaling through other guidance pathways. In an in vivo model of corneal neovascularization, Robo1 and 4 were upregulated, while Slit2 was downregulated, indirectly suggesting that Slit2/Robo4 signaling may play a role in the process of corneal lymphangiogenesis [127]. Although more research is necessary, the Slit2/Robo4 interaction serves as a potential therapeutic target for lymphatic dysfunction due to VEGF-C/VEGFR-3 activation.

### 3.8. Chemokines

Chemokines are known to be crucial factors for the generation of peripheral lymph nodes, which are draining sites for lymphatic vessels [128]. Chemokine receptors (CXCRs) and their ligands (CXCLs) serve as critical guidance cues that direct early lymphatic trunk assembly and network patterning in wound healing [129]. CXCR4 is expressed in the lymphatic endothelium, and CXCL12 is expressed in neighboring tissues into which lymphatic vessels migrate [70]. VEGF-C upregulates CXCR4 expression on LECs in a process mediated by hypoxia-inducible factor-1α (HIF-1α) [70]. However, this novel pathway also regulates lymphangiogenesis independent of the VEGFR-3 pathway. CXCL12 binds CXCR4 to induce migration and tube formation of LECs in vitro and lymphangiogenesis in vivo, and its expression level is correlated with lymphatic vessel density [70].

Deficiencies in atypical chemokine receptors such as atypical chemokine receptor 3 (ACKR3, formerly CXCR7) and ACKR2 are associated with hyperplasia and hyper-branched lymphatics [130,131]. Specifically, Kim et al. found that the inflammatory chemokine CXCL2 and its scavenging receptor, ACKR2, regulate lymphatic vessel density and the recruitment of pro-lymphangiogenic macrophages to areas around resting and regenerating lymphatic tissue [130]. ACKR2 was also observed to regulate the concentration of CXCL2 available near LEC surfaces and the concentration of pro-lymphangiogenic macrophages around developing LEC networks [130]. ACKR3 generates guidance cues for CXCL12-mediated endothelial cell migration; specifically, ACKR3b knockdown leads to loss of correct direction of migration and the presence of aberrant sprouts and ectopic filopodia protrusions [132].

Deletion of ACKR2 was also found to accelerate corneal lymphatic sprouting and LEC infiltration in the early stages following murine corneal transplantation but not to confer increased graft survival rates [133]. Following corneal neovascularization, administration of a CXCR4 antagonist decreases corneal lymphangiogenesis and VEGF-C/VEGFR-3 expression [134]. In contrast, CXCL10 expression decreases corneal lymphangiogenesis via VEGF-C and matrix metalloproteinase (MMP)-13 suppression [135]. These results illustrate the differential ability of chemokines to indirectly mediate lymphangiogenesis guidance in the cornea and inform the possibility of combination therapeutic approaches aimed at specific molecular targets known to regulate lymphangiogenesis.

### 3.9. Sphingosine 1-Phosphate

Sphingosine 1-phosphate (S1P) is a bioactive lipid that is an important mediator of inflammation and cancer via the regulation of cell proliferation, directional migration, and trafficking [136]. The absence of both S1P and its receptor, sphingosine 1-phosphate receptor 1 (S1PR1), has been shown to result in fatal vascular stabilization defects and to induce VEGF-C expression through an FGF-dependent pathway in human umbilical vein endothelial cells [71,72]. Interestingly, S1PR1 and S1P are also known to be key players in the directional migration of LECs in response to a fluid flow stimulus. For instance, S1PR1 is required for inward migration of LECs in response to wall shear stress and for persistent, directional migration in response to flow [73]. In addition, deficiency in S1P production in LECs affects the patterning of peripheral lymphatic vessels [76]. Particularly, lymphatic vessels show an irregular morphology, with reduced organization of VE-cadherin distribution at cell–cell junctions [76].

S1PR1 has been found to be expressed in human corneal fibroblasts [137], suggesting that the cornea could be used as a model for further studies of guidance by S1PR1. Application of topical S1PR1 was found to increase the duration of corneal graft survival following transplantation, but the underlying effect on LEC guidance in the cornea during this inflammatory state has yet to be understood [74].

### 3.10. Hyaluronan

Hyaluronan is a non-sulfated glycosaminoglycan component of the extracellular matrix (ECM) that modulates cell differentiation, proliferation, and growth. It is present in areas of cell division, inflammation, and ECM remodeling [138]. Low molecular weight hyaluronan promotes cooperation between LYVE-1 and S1P3 (a S1P receptor) at the cell surface to promote lymphangiogenesis [77].

Hyaluronan regulates organization of the lymphatic endothelium into vessel-like cell sheets, allowing for tube formation [78,138]. Sun et al. described an increase in hyaluronan levels followed by lymphatic vessel sprouting and growth in an alkali-burn injury model of the murine cornea [78]. Critically, lymphangiogenesis was only observed in regions with high hyaluronan concentrations. Given the spatiotemporal relationship between hyaluronan expression and lymphatic vessel patterning, these data suggest that hyaluronan is crucial to the formation and maintenance of corneal lymphatic vessels in both healthy and pathological states. Hyaluronan matrix-specific targets could yield a potential therapy for preventing pathological corneal lymphangiogenesis.

Interestingly, hyaluronan and VEGF-C administered together synergistically act to increase lymphatic tube formation and sprouting to a higher level than either hyaluronan or VEGF-C administered alone. Hyaluronan induces increased expression of VEGF-C from bone marrow-derived macrophages [139]. The findings of F4/80 macrophage-delineated vessels in late embryogenesis combined with hyaluronan’s ability to induce macrophage expression of VEGF-C suggest that hyaluronan could mediate the effects of macrophages in corneal lymphangiogenesis [8].

### 3.11. Integrin Family

The integrin family of ECM receptors couples lymphatic capillary endothelial cells to the surrounding matrix through force transduction to modulate LEC migration and proliferation. Several LEC integrins have been implicated in lymphangiogenesis, and multiple inhibitors that target α5β1, αvβ3, and α4β1 have been approved by the FDA or are in clinical trials [140].

Regarding guidance mechanisms, several integrins have been implicated in patterning and migration. α9β1 binds directly to VEGF-A/-C/-D, the fibronectin extra domain A (EDA), emilin1, and polydom. Knockout of this integrin results in defective lymphatic valve formation and inhibited LEC migration [79,80,81]. Integrin α9 has been shown to be involved in the developmental patterning of lymphatic vessels [141], and integrin α9 mutant mice show severe defects in lymphatic vessel sprouting [142]. Integrin α9 is also crucial in lymphatic valve morphogenesis, as its deletion causes rudimentary valve leaflets, disorganized fibronectin matrix, and cusps of retrograde lymphatic flow [79]. As explained previously, the proper maintenance of topographical cues, such as fibronectin, and flow mechanics are essential for the guidance of LECs. Integrin α9 seems to play a role in maintaining these processes. Additionally, integrin α6β1 mediates LEC adhesion and migration via the pro-lymphangiogenic factor netrin-4 [83].

Administration of an integrin α9-blocking antibody inhibits lymphatic valve formation post suture-induced corneal lymphangiogenesis [143]. Integrins have been shown to play a key role in the induction of corneal lymphangiogenesis via galectin-8, a carbohydrate-binding protein that stimulates pathological lymphangiogenesis [144]. Blockade of integrins α1β1 and α5β1 prevents LEC sprouting via galectin-8 and even VEGF-C, indicating their role in the promotion of lymphangiogenesis via multiple signaling pathways [145]. Specifically, integrin α1β1 inhibition results in decreased lymphatic vessel coverage and area of invasion [146]. Moreover, inhibition of α5 integrins prevents lymphatic vessel outgrowth during corneal inflammation post-injury while having a less pronounced effect on blood vessel outgrowth, highlighting integrin blockade as a potential method to selectively target corneal lymphangiogenesis in inflammatory settings [147].

### 3.12. BMP Family

Bone morphogenetic proteins (BMPs) are growth factors that regulate cell proliferation, differentiation, survival, and apoptosis [148]. BMPs have been implicated in lymphangiogenic guidance, but the exact mechanisms underlying this influence are unclear.

Niessen et al. found that the stimulation of activin receptor-like kinase 1 (ALK1) via ligands BMP9 and BMP10 is crucial to the development, maturation, and patterning of lymphatic vessels. Inhibition of ALK1 signaling leads to severe lymphatic developmental deficiencies in a plethora of organs and in developmental and tumorigenic processes [84]. Knockout of BMP9 results in hyperplastic lymphatic vessels, significant reductions in the number and maturation of lymphatic valves, and decreased lymphatic drainage capacity [149]. Another study showed that BMP9 directs early lymphatic endothelial tip cell expansion during embryonic stem cell differentiation in mice [150,151]. A low concentration of BMP9 enhances the LYVE-1-positive LEC lineage through ALK1 activation, whereas a high concentration of BMP9 enhances the LYVE-1-negative LEC lineage. At this high concentration, BMP9 also helps to activate VEGF-A which is a known potent guidance molecule for lymphangiogenesis [150]. Therefore, BMP9 seems to be involved in both the initiation process promoting the formation of LEC lineages and in the regulation process mediating the guidance of lymphatic vessels.

To elucidate the underlying signaling pathways through which BMPs contribute to corneal lymphangiogenesis, Hu et al. utilized a suture-induced injury model for rat corneal neovascularization [85]. They demonstrated an overall inhibitory effect of BMP4 on corneal lymphangiogenesis quantified by corneal neovascularization area and levels of lymphangiogenic factors. Smad is a major signaling protein that acts downstream of BMP4. Blockade of the BMP4/Smad pathway via the inhibitor Smurf1 alleviated the inhibitory effect of BMP4 on corneal neovascularization. Removal of exogenous BMP4 resulted in increased endogenous BMP4 levels, while expression of VEGF-C/-D and VEGFR-3 was inhibited, suggesting an inhibitory effect of endogenous BMP4 on lymphangiogenic growth factors. A delicate balance seems to exist between levels of exogenous BMP4, endogenous BMP4, and VEGF/VEGFR expression, underscoring the need to determine the specific mechanisms by which BMPs influence VEGF signaling.

### 3.13. Angiopoietin/Tie Receptors

Tie1 and Tie2 are endothelial receptor tyrosine kinases with corresponding angiopoietin (Ang) ligands. Ang-1/Tie2 signaling has been recognized for its pro-angiogenic role and was shown to promote vascular stability [152]. One study demonstrated that loss of Tie1 causes abnormal lymphatic patterning and disorganization of lymphatic vessels in examined tissues. In addition, Tie1 deficiency causes abnormal regression of lymphatic vasculature, highlighting its necessity in the survival of developing LECs [87]. Ang-2 has been shown to be required for angiogenic remodeling after embryonic development as well as lymphatic vessel patterning and function. Specifically, Ang-2 was shown to guide the regular sprouting of lymphatics around arteries and veins, suggesting a relationship between the guidance of lymphatics and blood vessels [88]. In addition, intestinal and skin lymphatics in Ang2-LacZ (β-galactosidase) knockout allele mice were shown to be disorganized and irregular, in part, due to a lack of coordinated migration [88].

Interestingly, Ang and Tie2 signaling is critical for maintenance of the integrity of the Schlemm’s canal (SC), a circular vessel in the eye that delivers collected aqueous humor to the episcleral blood vessels. Due to its function in fluid homeostasis, the endothelium lining the inner wall of the Schlemm’s canal has prominent lymphatic vasculature, which is influenced by lymphatic guidance cues [153]. Both pericyte-derived Ang1 and endothelium and trabecular meshwork-derived Ang2 play a role in this pathway. Upon deletion of Tie2 and Ang1/Ang2, the Schlemm’s canal faced severe regression and elevated intraocular pressure, a major cause of glaucoma. However, deletion of Ang1 or Ang2 reduced cellularity and levels of Prox1 and Klf4, which are shear stress-responsive transcription factors, but did not reduce cellularity and levels of Tie2 in the Schlemm’s canal [154]. The Schlemm’s canal area was only mildly reduced when only Ang2, was deleted, implying that Ang1 and Ang2 have related, yet distinct roles in SC maintenance. Additionally, Ang2 was downregulated in patients with primary open-angle glaucoma, indicating that Ang2 plays an agonistic role compared to Tie2 in the development and maintenance of the Schlemm’s canal [154].

A nearly complete inhibition of inflammatory lymphangiogenesis is observed in mouse corneas following Ang-2 knockout, suggesting it plays an essential role in corneal lymphatic development [26]. Comparatively, Ang-2 knockout results in disruption of the blood vessel network without complete abolition, highlighting the increased susceptibility of lymphangiogenesis to treatments targeting Ang-2. In the same study, subconjunctival treatment with Ang-2 siRNA blocked Ang-2 signaling and suppressed corneal lymphangiogenesis post-injury. Finally, Ang-2 has been shown to sensitize LECs to inflammatory stimuli during corneal injury in addition to its pro-lymphangiogenic properties [155].

### 3.14. TGFBIp

Transforming growth factor-beta-induced protein (TGFBIp) promotes sprouting via the formation of lymphatic endothelial tips in the cornea but is unable to stimulate inward growth of lymphatic vessels [89]. When administered with VEGF-C, however, the two synergistically upregulate corneal lymphangiogenesis, with TGFBIp-mediated conversion of LECs to a tip cell phenotype guiding VEGF-C-mediated lymphatic vessel extension. Suppression of lymphatic vessel sprouting via blockade of integrin α5β1 indicates that TGFBIp–integrin α5β1 binding and subsequent induction of sprouting represents a potential mechanism of initial lymphatic vessel germination.

### 3.15. Semaphorins

Class 3 semaphorins (Sema3A-H) were initially described as axonal growth cone guidance molecules but have recently been identified as key players in repulsive lymphangiogenic and vascular endothelial cell guidance [93,156]. For example, Sema3A–NRP1–plexinA1 signaling is involved in lymphatic valve formation and vessel function [94,95]. Semaphorins can provide bidirectional guidance, acting either as repulsive or attractive guidance molecules to LECs. Repulsive guidance negatively impacts the function of LECs and mediates repulsion of endothelial cell migration. Additionally, repulsion can also mediate the natural process in which LECs are repelled from blood vessels to form random lymphatic vascular networks in peripheral tissues.

Uchida et al. revealed that Sema3F and Sema3G negatively regulate lymphatic sprouting and proliferation to control lymphatic cell patterning [157]. Sema3F is expressed in the epidermis, and Sema3G displays selective expression in arteries. Their receptor Nrp2 is preferentially expressed by lymphatic vessels. Sema3F and Sema3G mutants demonstrate different phenotypes in lymphatic branching and LEC growth. Sema3F was found to globally influence LEC sprouting and proliferation at low concentrations, while Sema3G was found to locally inhibit lymphatic sprouting at relatively high concentrations [157]. Sema3F and Sema3G also display different binding affinities for plexin complexes. Sema3F signaling functions specifically via the NRP2/plexinA3 or NRP2/plexinA1 complexes, while Sema3G signaling functions preferentially via the NRP2/plexinA2 complex. The class 7 semaphorin Sema7A activates the β1-integrin receptor to stimulate lymphangiogenesis, highlighting the differential roles of semaphorins in the regulation of lymphangiogenesis [158].

Genetic inactivation of Sema3G or plexinD1 in mouse embryonic back skin results in abnormal artery–lymph alignment and reduced lymphatic vascular branching, highlighting Sema3G’s role in distributing LECs away from arteries to form a free branching network in an NRP-2/plexinD1-dependent manner [101]. Additionally, in a model of head and neck squamous cell carcinoma, genomic loss of Sema3F correlated with increased metastasis and decreased survival, suggesting that Sema3F promotes LEC collapse and inhibits lymphangiogenesis in vivo [98]. Re-expression of Sema3F decreased lymphangiogenesis and lymph node metastasis. Sema3F overexpression was shown to have a direct chemorepulsive effect on LECs [99] via competition with VEGF-C for NRP-2 binding [100].

Reuer et al. described marked downregulation of Sema3F expression during inflammatory states in a mouse model of suture-induced corneal lymphangiogenesis [159]. The application of topical Sema3F eye drops was found to inhibit outgrowth of the corneal lymphatic network without impacting blood vessels following injury, identifying Sema3F as a potential target for selective inhibition of lymphangiogenesis following ocular procedures. During murine corneal transplantation, treatment with Sema3F eye drops during the prevascularization phase decreases graft rejection rates. These beneficial findings for Sema3F administration highlight the therapeutic modality of promoting anti-lymphangiogenic corneal guidance pathways. Future research must compare the efficacy of downregulating pro-lymphangiogenic pathways and upregulating anti-lymphangiogenic pathways in the cornea, as well as the potential synergistic effect of a combinatory approach.

### 3.16. Notch Proteins

Signaling between the Notch receptor and its ligand Delta-like 4 (Dll4) regulates the determination of cell fate [160]. Upon overactivation of the Notch1 protein during embryonic development, Prox1-expressing LECs are suppressed and peripheral lymphatic vessels abnormally contain blood endothelial cells [161]. Zheng et al. demonstrated that in response to both VEGF-A and VEGF-C signaling, Dll4 expression in LECs is upregulated, which activates Notch signaling and suppresses lymphangiogenesis [32]. Furthermore, suppression of Notch signaling in vivo by Dll4-Fc promotes lymphangiogenesis. Interestingly, lymphangiogenesis was augmented by the provision of VEGF-A more than by the provision of VEGF-C, suggesting that VEGFR-2 signaling is inhibited to a greater extent by Notch signaling than is VEGFR-3 signaling. In another study, Notch1 mutant mice exhibited excessive lymphatic sprouting and dilated lymphatic vessels during embryogenesis, highlighting the important role that Notch signaling plays in regulating VEGF-C/VEGFR-3-induced lymphatic sprouting [44]. Noncanonical Notch4 signaling has recently been implicated in lymphangiogenesis regulation, as LEC migration is inhibited more severely by Notch4 activation than by Notch1 activation [162].

A potential role for Notch signaling in VEGF-A-mediated lymphangiogenesis was recently uncovered in a bFGF-induced murine model of corneal lymphangiogenesis. Xie et al. showed increased mRNA and protein expression of Notch1 in mouse corneas after implantation of bFGF pellets [163].

### 3.17. Other Guidance Molecules

In addition to the aforementioned guidance cues, several novel endogenous regulators of lymphangiogenesis have been identified in the cornea. The specific role of each of these factors in the guidance of corneal lymphangiogenesis has yet to be determined.

Several studies have provided evidence that microRNAs (miRNAs) play roles in mediating guidance cues for directing lymphatic budding. Chen et al. showed that in zebrafish, miR-126a is crucial for parachordal lymphangioblast extension along the horizontal myoseptum by positively regulating the expression of CXCL12a, suggesting that miR-126a plays a role in the directional guidance of lymphatic development [164]. miR-466 was shown to act as a negative regulator of Prox1 expression following alkali burn-induced corneal lymphangiogenesis [165]. In a rat model of alkali burn corneal injury, injection of miR-199a/b-5p suppresses corneal lymphangiogenesis by downregulating discoidin domain receptor 1 (DDR1) [166].

Blockade of the matrix metalloproteinases MMP-2 and MMP-9 prevents corneal lymphangiogenesis via decreased macrophage recruitment and VEGF-C expression, highlighting the role of ECM degradation in lymphangiogenesis guidance [167]. Conversely, membrane type 1-MMP (MT1-MMP) specifically suppresses corneal lymphangiogenesis by cleaving LYVE-1 on LECs, suggesting an anti-lymphangiogenic role [168].

In embryonic development, the extracellular protease ADAMTS3 (a disintegrin and metalloproteinase with thrombospondin motifs) stimulates lymphangiogenesis by cleaving pro-VEGF-C into the active VEGF-C, facilitating VEGF-C/VEGFR-3 signaling [169]. Recently, it was shown that knockout of the structurally similar ADAMTS2 and ADAMTS14 in mice results in decreases in the number of lymphatic vessels, vessel length, and branching [170]. Interestingly, ADAMTS2^−/−^ ADAMTS14^−/−^ double knockouts showed amplification of these inhibitory effects in the same study, suggesting a collaborative effect of both proteases during inflammatory lymphangiogenesis. CCBE1 (collagen- and calcium-binding EGF domain-containing protein 1) is an essential protein for lymphangiogenesis that enhances the cleavage of pro-VEGF-C by ADAMTS3, facilitating lymphatic vessel development [171].

Several other regulators of lymphangiogenesis guidance have been described. A protein interaction analyses of 43 guidance molecules elucidated four potential master regulators of corneal lymphangiogenesis (Figure 4).

## 4. Pathologies Involving Lymphatic Endothelial Guidance

### 4.1. Dry Eye Disease

Dry eye disease is an ocular surface disease caused by tear film disruption and inflammation of the ocular surface that manifests as redness, dryness, discomfort, and light sensitivity [12]. In recent years, the ingrowth of corneal lymphatic vessels that facilitate immune cell trafficking in the afferent arm has been identified as critical to understanding the disruption of ocular surface homeostasis [172]. The production of pro-lymphangiogenic factors has been shown to drive the invasion of lymphatic vessels into the cornea, impairing lymphangiogenic privilege and impairing visual acuity.

Critically, corneal lymphangiogenesis in dry eye disease is not accompanied by hemangiogenesis, underscoring the importance of understanding and targeting lymphatic endothelial guidance in dry eye disease pathogenesis [113]. Desiccating stress causes corneal lymphangiogenesis from the limbal arcades and increased expression of VEGF-D and VEGFR-3 early in the disease course. Comparatively, promotion of VEGF-A, VEGF-C, and VEGFR-2 is delayed [113]. This differential expression with respect to disease progression highlights the initial role of VEGF-D in inducing LEC guidance and its ability to modulate the migratory and sprouting effects of VEGF-C-mediated guidance later in the disease [173].

Binding of the neuropeptide substance P (SP) to the neurokinin 1 receptor (NK1R) induces lymphatic tube formation and proliferation in vitro through upregulation of VEGF/VEGFR-3 signaling [174]. Administration of NK1R antagonists reduces corneal lymphangiogenesis and critically inhibits the direction of lymphatic vessel patterning towards the center of the cornea in dry eye disease and alkali burn murine models [174,175]. This blockade of SP/NK1R signaling also results in a decrease in VEGF-C/-D/VEGFR signaling. Symptomatically, NK1R antagonism can alleviate punctate damage to the epithelium and decreases in tear volume [174]. The suppression of corneal lymphangiogenesis via NK1R antagonism in dry eye disease sheds light on a potential repellant guidance mechanism involving disruption of SP/NK1R signaling.

Thrombospondin-1 (TSP-1), a multifunctional ECM glycoprotein expressed in humans, has been shown to both prevent LEC tube formation in vitro and play an inhibitory role against lymphangiogenesis in dry eye disease [176]. TSP-1 knockout mice exhibit increased corneal lymphangiogenesis and corneal VEGF-C expression [177]. The proposed mechanism underlying this inhibition involves the binding of TSP-1 to CD36 on cells from the monocyte lineage, which subsequently downregulates VEGF-C expression in these cells and reduces corneal lymphatic vessel area [178]. Critical to the initiation of immunopathogenesis in dry eye disease is the proinflammatory cytokine interleukin (IL)-17, which facilitates VEGF-C/-D/VEGFR-3 interactions and promotes corneal lymphangiogenesis [176].

### 4.2. Corneal Graft Rejection

Due to the avascularity of the cornea, corneal transplantation is associated with lower rejection rates than similarly performed allografts of other human tissues, a phenomenon known as “immune privilege” [7]. Despite this immune privilege, corneal transplant rejection still occurs. A major inflammatory event underlying corneal graft rejection is the ingrowth of pathologic blood and lymphatic vessels from the limbal arcades into the previously avascular cornea (Figure 5) [179]. Lymph vessel invasion into the cornea facilitates antigen presentation to regional lymph nodes and sensitization to donor tissue, while blood vessel invasion facilitates the influx of effector T cells to the site of transplantation, thereby promoting graft rejection [180]. It has been shown that prevascularization of corneal beds is correlated with increased corneal graft rejection rates [181]. Whereas the role of hemangiogenesis in corneal graft rejection has been well studied and serves as the basis of many approved therapeutics, knowledge of the effects and mechanisms of lymphangiogenesis and specifically lymphatic vessel guidance remains premature in comparison.

The (lymph)angiogenic privilege of the cornea was shown to contribute to lower transplant rejection rates compared to those for vascularized organs specifically [182]. However, Dietrich et al. showed that corneal transplantation into corneal beds containing blood vessels but not lymphatic vessels is associated with a greater graft survival rate than transplantation into corneal beds containing both vessel types [183]. This finding suggests that the presence of lymphatic vessels prior to transplantation confers a high-risk status to recipients.

Therapeutics targeting corneal lymphangiogenesis specifically or neovascularization in general have both shown success in decreasing corneal graft rejection rates. Despite the relatively minor role of VEGF-A in lymphangiogenesis promotion compared to VEGF-C, neutralization of VEGF-A by VEGF Trap (R1R2) and bevacizumab (anti-VEGF-A monoclonal antibody) have both been shown to improve graft survival [184,185]. Administration of antibodies against VEGF-C and VEGFR-3 also specifically inhibits lymphangiogenesis and promotes graft survival [183,186].

Soluble VEGFR-2 (sVEGFR-2) has been identified as an endogenous suppressor of lymphangiogenesis in the cornea via blockade of VEGF-C signaling, and corneal treatment with VEGFR-2 decreases graft rejection [187]. Endogenous soluble VEGFR-3 is critical for maintenance of corneal lymphangiogenic privilege, and its overexpression heavily increases graft survival rates during corneal transplantation via VEGF-C/VEGFR-3 blockade [188].

Apart from VEGF signaling, the administration of a monoclonal antibody against integrin α9 inhibits the formation of lymphatic vessels and promotes corneal graft survival [189]. As previously mentioned, topical treatment with the inhibitory guidance molecule Sema3F limits corneal lymphangiogenesis and subsequently increases corneal transplant survival rates [159]. Ang-2-neutralizing antibody treatment similarly promotes graft survival and inhibits corneal lymphangiogenesis, preventing trafficking of donor cells to regional lymph nodes [190].

Maruyama et al. described the beneficial effects of inhibiting podoplanin, a lymphatic endothelial-specific marker, on corneal transplant survival via administration of an anti-podoplanin monoclonal antibody [191]. These results highlight novel guidance pathways outside of the VEGF/VEGFR signaling cascade as potential targets for regulating corneal lymphangiogenesis in corneal transplantation. The identification of lymphatic-specific molecular players in graft rejection paves the way for anti-lymphangiogenesis-specific treatment paradigms targeting the afferent arm of the immune response.

### 4.3. Infectious Keratitis

Infectious keratitis is a leading cause of blindness from infectious etiologies worldwide and most commonly involves Herpes simplex virus 1 (HSV-1). This disease is known as herpetic stromal keratitis (HSK) [192]. HSV-1 is estimated to be prevalent in around two-thirds of the global population and can take on a myriad of virulence patterns depending on characteristics of the host that it infects [193]. Specifically, the relationship between HSV-1 and the host’s immune system serves as a crucial predictor of disease pathogenesis [193]. Similar to dry eye disease, HSV-1 infection induces pathological corneal lymphangiogenesis and compromises immune privilege [11].

Upon the development of lymphatic vessel-specific markers such as LYVE-1, initial visualization of HSK illustrated lymphatic vessel invasion of corneal tissue starting as early as the first day following infection and continuing for the duration of one week [111]. In terms of initial pathogenesis, VEGF-A is implicated as the main VEGF ligand that drives lymphatic vessel growth in contrast to its accessory prolymphangiogenic effects in physiological states [111]. HSV-1 indirectly promotes lymphangiogenesis by recruiting MMP-7 to degrade sVEGFR-1, downregulating mRNA expression of the VEGF-A trap sVR-1 and destroying existing sVR-1 [187,194]. In the setting of HSK, the role of VEGF-C from CD8+ T cells is to induce the growth of preexisting lymphatic vessels at a later point in time than VEGF-A [195]. At later stages of infection, CD4+ T cells and infiltrating neutrophils begin to mediate immunopathogenesis [196].

Induction of lymphangiogenesis in chronic HSK is responsible for impairment of visual acuity and ultimately blindness by providing an avenue for several deleterious inflammatory molecules [196]. Thus, several VEGF-A antagonists, such as bevacizumab, have been applied to HSK in an attempt to stifle lymphangiogenesis at the earliest possible stage [197]. Considering the critical role of VEGF-A/VEGFR-2 signaling on HSK pathogenesis, administration of soluble VEGFR-2 is a useful strategy for the selective inhibition of corneal lymphangiogenesis while leaving hemangiogenesis intact [187]. Choudhary et al. demonstrated a complete reduction of the immunomodulatory molecule TSP-1 within 24 h following HSV1 infection in vitro, suggesting a potential therapeutic modality of recombinant TSP-1 during the early stages of HSK [198]. Finally, insulin receptor substrate-1 (IRS-1) has been implicated in lymphangiogenesis in the setting of HSK; treatment with GS-101, an antisense oligonucleotide that inhibits IRS-1, inhibits corneal lymphangiogenesis [199,200].

As mentioned previously, in a study of bacterial keratitis, corneal lymphangiogenesis and levels of VEGF-C/VEGFR-3 were increased in the later stages of the disease, a result that did not extend to corneal hemangiogenesis [15]. Interestingly, enhanced levels of corneal lymphangiogenesis reduced corneal edema and improved the clinical course of the disease at this stage [15].

### 4.4. Alkali Burn

Alkali burns are a type of ocular chemical injury that can result in damage to the cornea by causing changes in pH, ulceration, proteolysis, and defects in collagen synthesis [201]. On account of their lipophilic nature, alkali compounds rapidly penetrate the cornea, damage tissue via saponification, facilitate the release of proteolytic enzymes, and subsequently stimulate the inflammatory response [201]. The degree of correlation between corneal lymphangiogenesis and corneal inflammation is strong in rat models of alkali-burned corneas, highlighting the critical role of corneal lymphatic vessels in forming the afferent limb of the inflammatory response following alkali burn [202].

On account of its immunostimulatory properties, alkali burn is often used to produce corneal injury models for investigating the impact of gene knockout and various protein compounds on pathological corneal lymphangiogenesis [165,203,204,205]. When used in combination with dual-transgenic imaging techniques, the corneal alkali injury murine model allows for simultaneous live visualization of blood and lymphatic vessel growth [206].

Different methods of corneal injury can induce vastly different inflammatory profiles including various degrees of lymphangiogenesis [207]. Corneal alkali burns induce a relatively modest lymphangiogenic response compared to keratoplasty and suture placement but a stronger response than incision [207]. In a direct comparison between suture-induced and alkali burn-induced corneal inflammation, alkali burn produced a less pronounced lymphangiogenic response but a similar hemangiogenic response, indicating that differences in the method of induction of inflammation may be significant for corneal lymphangiogenesis [208]. Zhu et al. demonstrated the induction of more robust lymphangiogenesis upon corneal alkali burn compared to VEGF or bFGF pellet implantation into the cornea [209]. Initial growth of lymphatic vessels occurred at day 3 post-injury in all directions and growth peaked between days 3 and 7 post-injury [209]. In contrast, Ling et al. show that corneal lymphatic vessels peak two weeks following alkali injury and regress faster than blood vessels in a corneal injury murine model [210].

## 5. Therapeutic Strategies Targeting Lymphangiogenesis Guidance Mechanisms

Preclinical models of corneal lymphangiogenesis have revealed novel molecular pathways not previously thought to be involved in guidance. Many findings from these models provide a broader picture of VEGF/VEGFR signaling as a primary regulator of lymphangiogenesis through which other molecular pathways interact to impact vascular network patterning [18]. Several novel exogenous agents have been shown to influence corneal lymphangiogenesis following injury (Table 2). The specific underlying mechanisms by which these novel agents influence lymphangiogenesis have yet to be understood, and such knowledge will facilitate the development of unique combinatory approaches targeting multiple signaling pathways that regulate corneal lymphangiogenesis. Various endogenous and exogenous guidance cues can disrupt the delicate balance between pro- and anti-lymphangiogenic factors, either inducing or suppressing corneal lymphangiogenesis (Figure 6).

Despite the wealth of active and completed clinical trials investigating lymphangiogenic guidance molecules currently described in the literature, many of these molecules have been investigated in contexts unrelated to corneal lymphangiogenesis (Table 3). For example, although Ang-2 has been implicated in corneal lymphangiogenic sprouting, Ang-2-modulating agents have historically been tested only for their effects on hemangiogenesis in clinical trials [219]. Even VEGF-C/VEGFR-3 signaling, the major pathway in lymphatic vessel patterning, has only recently been investigated. Few trials mention the role of lymphangiogenic guidance cues in impacting lymphatic vessel fate, with even fewer including the effect on lymphatics as a primary outcome measure. Additionally, despite the success of several anti-angiogenic agents in the cornea, there has been a lack of emphasis on translating well-characterized preclinical lymphangiogenic guidance inhibitors identified in the cornea to clinical trials. To date, there exists no completed clinical trials investigating corneal lymphatic vessel guidance as an outcome of drug treatment. There is a need for increased emphasis on recapitulating the beneficial results of murine corneal lymphangiogenesis inhibition in humans and the incorporation of lymphatic vessel parameters as a primary outcome measure in drug studies involving lymphangiogenic guidance molecules.

## 6. Occurrence of Lymphangiogenesis without Hemangiogenesis

In normal human anatomy, lymphatic vessels are closely associated with blood vessel networks, necessitating a discussion on blood vessel-mediated guidance of lymphangiogenesis. In zebrafish, cells resembling LECs migrate along the mesencephalic vein during development, remaining close to blood vessels in the meninges throughout adulthood [220,221]. Alterations in blood vessel patterning during zebrafish development impact the resulting lymphatic vessel network, indicating their role in guiding the initial stages of lymphangiogenesis [222]. Adrenomedullin is a pro-lymphangiogenic guidance factor regulated by CXCR7, an atypical chemokine receptor. Deletion of CXCR7 promotes increased lymphatic vessel sprouting during development [131]. CXCR7 is primarily expressed on blood endothelial cells, implicating the role of blood vessels in impacting the guidance of lymphangiogenesis.

Upon disruption of corneal privilege via inflammatory insult, blood vessels and lymphatic vessels invade the cornea in close spatiotemporal proximity to each other [209]. While in some instances hemangiogenesis can guide lymphangiogenesis, several sources have described the growth of lymphatic vessels independent of blood vessel growth. Zhong et al. observed lymphatic vessels without corresponding blood vessels in a Prox-1-GFP/Flt1-DsRed fluorescent mouse reporter model of VEGF-C corneal pellet implantation [206].

In corneal graft rejection, differences in hemangiogenesis and lymphangiogenesis in infants and adults provide a partial explanation for the inverse relationship between patient age and risk of corneal graft rejection [223,224]. Analysis of vessel diameter shows that a disparity exists between infant and adult lymphangiogenesis levels that is not observed between infant and adult hemangiogenesis.

Upon observation of the earlier appearance and longer lifetime of blood vessels compared to lymphatic vessels in mice, the presence of a preexisting blood vessel network was thought to be a requirement for lymphatic vessel guidance [225]. However, in the murine cornea, evidence of lymphangiogenesis in the absence of hemangiogenesis has been recently described in corneal injury models [113,226]. Despite regression of hemangiogenesis and VEGF-A levels, bFGF corneal pellet implantation results in increased levels of VEGF-C/-D and VEGFR-3 in a murine model of corneal lymphangiogenesis [227].

These differences illustrate potential mechanisms by which lymphangiogenesis differs from hemangiogenesis. Dual-encoded fluorescent studies of vascular network development and injury-induced vessel proliferation will help further elucidate the interactions between blood vessels and lymphatic vessels and the developmental checkpoints that separate their fates.

## 7. Conclusions

In this review, we provided an overview of lymphangiogenesis guidance cues, the underlying mechanisms through which they interact with lymphatic vessels in the cornea, their roles in corneal pathologies, and potential therapeutic approaches for regulating corneal lymphangiogenesis. While corneal blood vessel guidance has been well described in the literature, the role of lymphangiogenesis guidance in the cornea remains understudied, despite its critical involvement in initiating the immune response as part of the afferent arm of the inflammatory response to corneal injury. The observation that some guidance cues primarily act at specific checkpoints along lymphatic vessel development provides an avenue for alternative and combinatory approaches to regulate lymphangiogenesis. The translation of these preclinical findings into clinical trials is crucial. Several modulators of lymphangiogenic pathways have already been tested for various other purposes in clinical trials. Repurposing of these agents within the context of corneal lymphangiogenesis could provide an efficient and fruitful method for developing efficacious therapeutics.

## Figures and Tables

**Figure 1 cells-12-00319-f001:**
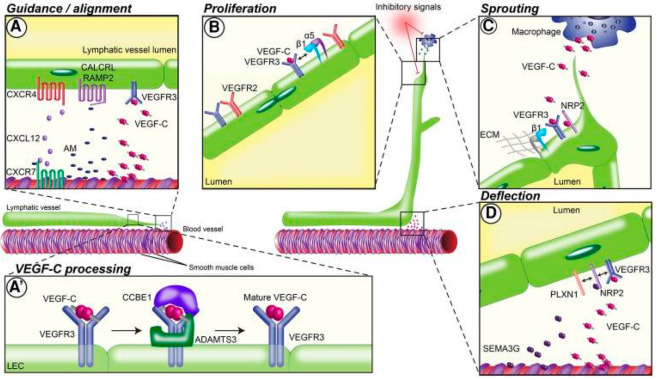
Guidance mechanisms for lymphatic network patterning in different stages of lymphatic vessel development. (**A**) Preexisting lymphatic vessels align with nearby blood vessels by reacting to guidance molecules released by arteries and smooth muscle cells. Guidance at this stage is mediated by the molecules VEGF-C and CXCL12 and the receptors VEGFR-3, CXCR4, CXCR7, CALCRL, and RAMP2. (**A’**) Vascular endothelial growth factor (VEGF)-C binds to surface proteins on the lymphatic endothelial cell (LEC) matrix and is processed into mature VEGF-C. VEGF-C processing is primarily mediated by CCBE1 and ADAMTS13. (**B**) VEGF-C binding to VEGF receptor (VEGFR)-2/3 and interaction with integrins induces LEC proliferation; α5 and β1 integrins also play a role at this step. (**C**) Macrophages release additional guidance molecules that initiate lymphatic vessel sprouting and branching. VEGF-C binding to its receptor VEGFR-3 and the coreceptor NRP2 drives this step, along with β1 integrins. (**D**) Deflection of lymphatic vessels during network formation occurs via release of repulsive guidance cues such as SEMA3G from nearby blood vessels. VEGFR-3, NRP2, and PLXN1 form a receptor complex that binds VEGF-C [18]. Reproduced with permission.

**Figure 3 cells-12-00319-f003:**
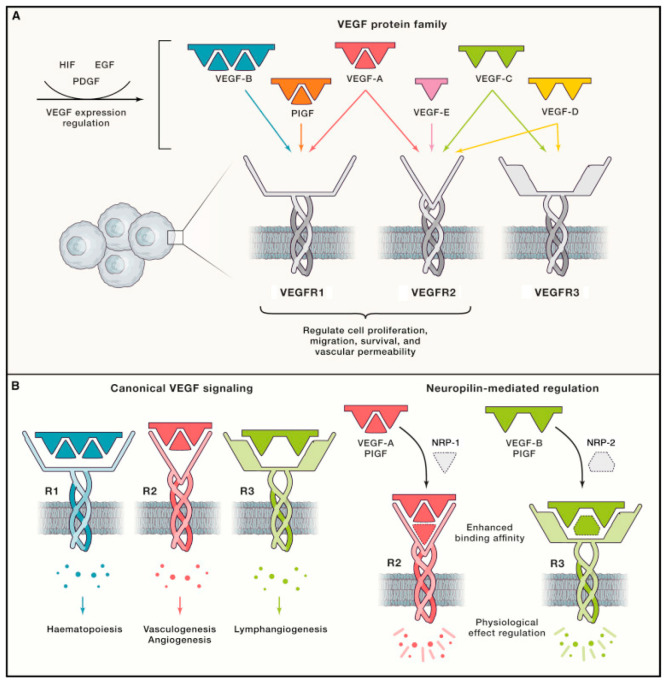
Schematic diagram of interactions between VEGFs and VEGFRs known to mediate hemangiogenesis and lymphangiogenesis. (**A**) The five major VEGFs expressed in mammals identified to date include VEGF-A to -D and placental growth factor (PlGF). VEGF-E is virally encoded. Each VEGF subtype interacts with either one or two of the three known VEGFRs to influence cell proliferation, migration, survival, and vascular permeability. VEGF expression is regulated by various factors, including hypoxia-inducible factor (HIF), epidermal growth factor (EGF), and platelet-derived growth factor (PDGF). (**B**) Canonical VEGF–VEGFR signaling mediates vasculogenesis, hemangiogenesis, lymphangiogenesis, and hematopoiesis. Vascular endothelial cells express VEGFR-1 and -2, and VEGF-A/VEGFR-2 signaling is the primary driver of hemangiogenesis. LECs express VEGFR-2 and -3, and VEGF-C and -D/VEGFR-3 interactions are the main drivers of lymphangiogenesis. Neuropilin-1 and -2 (NRP-1/-2) act as coreceptors for VEGFR-2 and -3, respectively, increasing the affinity of VEGF–VEGFR binding [104]. Reproduced with permission.

**Figure 4 cells-12-00319-f004:**
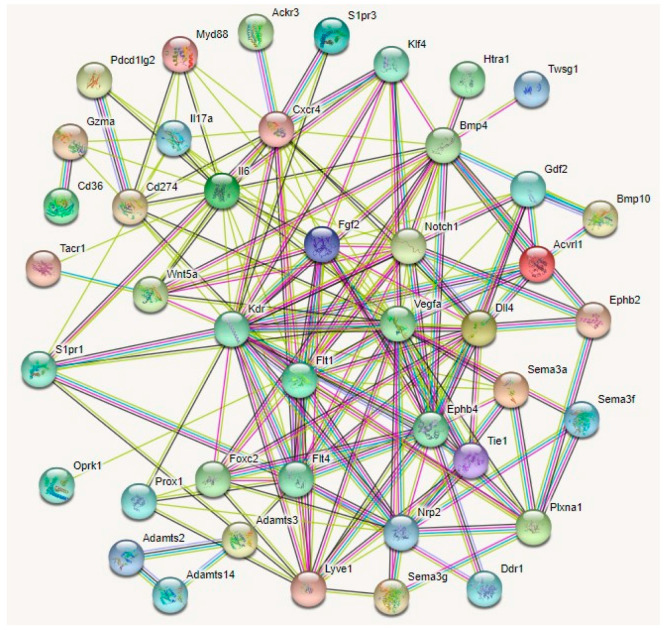
Protein–protein interaction networks for 43 lymphangiogenesis guidance molecules by STRING (Search Tool for the Retrieval of Interacting Genes/Proteins) analysis. By direct/indirect interactions with other guidance molecules Notch1, Nrp2, Fgf2, Vegfa, Kdr, Flt1, Ephb4, Flt4, and FOXC2 may act as master regulators mediating corneal lymphangiogenesis, providing potential avenues for the development of anti-lymphangiogenic therapies for corneal inflammatory pathologies including dry eye disease and cornea graft rejection.

**Figure 5 cells-12-00319-f005:**
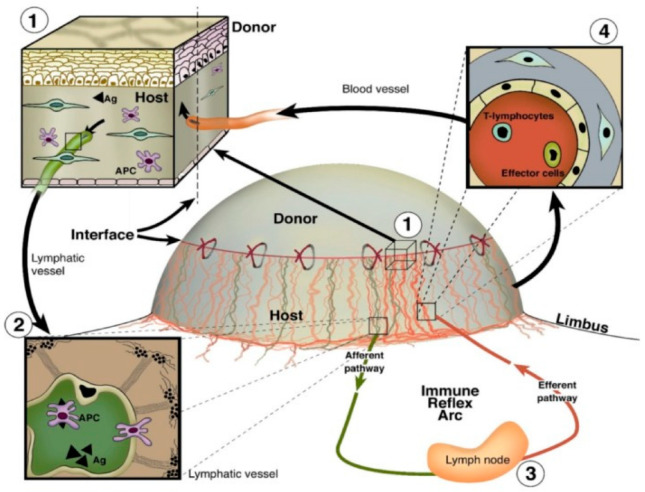
Blood and lymphatic vessel function in the afferent and efferent arms of inflammatory corneal neovascularization following corneal transplantation. (**1**) Antigens and antigen-presenting cells (APCs) surround the transplant surface (**2**) and enter afferent lymphatic vessels to migrate towards regional lymph nodes. (**3**) Activation of T lymphocytes and effector cells occurs (**4**), and these cells are transported back to the site of injury via efferent blood vessels [179]. Reproduced with permission.

**Figure 6 cells-12-00319-f006:**
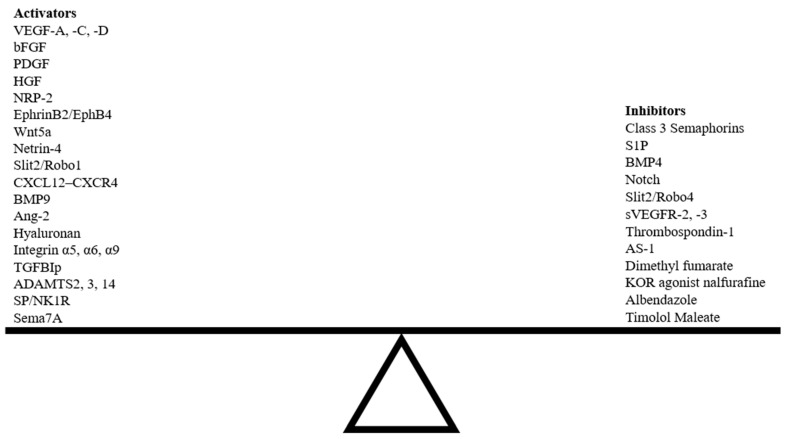
Balance between activators and inhibitors of corneal lymphangiogenesis guidance pathways. Abbreviations; HGF, hepatocyte growth factor; ADAMTS, a disintegrin and metalloproteinase with thrombospondin motifs.

**Table 2 cells-12-00319-t002:** Novel exogenous regulators of lymphangiogenesis guidance in the cornea.

Molecule	Signaling Pathway Targeted	Mechanism of Action	Findings	Reference(s)
AS1	IL-1β/IL1R1/ MyD88/NF-κB	A synthetic Toll/IL-1 receptor (TIR)/BB-loop mimetic that prohibits interaction between IL-1RI and MyD88	AS1 treatment decreases corneal lymphatic vessel ingrowth, VEGF-A/-C and LYVE-1 expression, and proinflammatory cytokine levels. AS1 alleviates corneal opacity, edema, and inflammatory cell infiltration post-alkali burn injury.	[211]
Nalfurafine	Kappa opioid receptor (KOR) signaling	KOR agonism	Nalfurafine treatment suppresses corneal lymphangiogenesis and VEGF-A/-C expression.	[212]
Dimethyl fumarate (DMF)	Cytokine-mediated macrophage signaling (TNF-α, IL-6, IL-1β, and VEGF-C)	Inhibition of NF-κB pathway activation in macrophages	Subconjunctival DMF injections decrease CD11b+ macrophage infiltration into the cornea and reduce mRNA expression of various proinflammatory cytokines (IL-1β, IL-6, TNF-α, MCP-1) following mouse corneal transplantation. As a result, DMF treatment inhibits macrophage-induced corneal lymphangiogenesis and decreases corneal graft rejection.	[213]
Topical VEGF-C/D Trap	VEGF-C/D/ VEGFR-3	VEGF Trap (sVEGFR3 + Fc portion) binds to VEGF-C/-D and blocks activity	VEGF-C/-D Trap reduced lymphangiogenesis in a suture-induced corneal injury mouse model while increasing the frequency of CD11b+ macrophages. No benefit was observed for corneal graft survival.	[214]
Albendazole	VEGF/VEGFR + TNF-α	Inhibition of VEGF transcription	Treatment with albendazole inhibits corneal lymphangiogenesis and downregulates VEGF-A/-C and TNF-α expression in a suture-induced corneal injury mouse model. Combination treatment with bevacizumab offers an additive effect on lymphangiogenesis reduction.	[215,216]
Timolol maleate	β adrenergic receptors	Nonselective blockage of β adrenergic receptors	Treatment with timolol maleate inhibits corneal lymphangiogenesis, VEGF-A/-C and VEGFR-2/-3 expression, and inflammatory cell infiltration in a suture-induced corneal injury mouse model.	[217,218]

Abbreviations: AS1, hydrocinnamoyl-L-valylpyrrolidine; IL-1β, interleukin-1 beta; IL1R1, interleukin 1 receptor 1; MyD88, myeloid differentiation primary response gene 88; NF-κB, nuclear factor kappa B; TNF-α, tumor necrosis factor alpha; MCP-1, monocyte chemoattractant protein-1; CD11b+, cluster of differentiation 11b+.

**Table 3 cells-12-00319-t003:** Clinical trials involving modulators of lymphangiogenesis guidance pathways used in the cornea.

Guidance Cue	Reference/ID	Agent Name	Mechanism of Action	Indication/Disease	Trial Phase
VEGF	NCT01072357	Bevacizumab	Anti-VEGF-A mAb	Corneal neovascularization Corneal graft failure	Phase 1/2 completed
	NCT01868360	Aflibercept	VEGF Trap (recombinant fusion protein)	Corneal neovascularization	Terminated
	NCT02342392	Ranibizumab	VEGFR-1,-2,-3 inhibitor	Pterygium	Phase 2/3 completed
Hyaluronan	NCT00599716	Vismed	Contains sodium hyaluronate	Dry eye disease	Phase 3 completed
	NCT01387620	Hyaluronic acid	Source of hyaluronan	Corneal edema	Phase 4 completed
	NCT05313425	Ectohylo eye drops	Contains sodium hyaluronate	Corneal haze Photorefractive keratectomy	Recruiting

Abbreviations: mAb, monoclonal antibody.

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
