# Peer review of "Lymphangiogenesis Guidance Mechanisms and Therapeutic Implications in Pathological States of the Cornea"

_cells, 2023, doi:10.3390/cells12020319_

Round 1
Reviewer 1 Report
This is a very well-researched and well-written article and gives an excellent overview of the state of the art in understanding lymphangiogenesis. It will be of tremendous utility to those in the field of angiogenesis or corneal research.
Author Response
"This is a very well-researched and well-written article and gives an excellent overview of the state of the art in understanding lymphangiogenesis. It will be of tremendous utility to those in the field of angiogenesis or corneal research."
Response: We thank the reviewer for their careful consideration of the manuscript and for recognition of our proposed manuscript.
Reviewer 2 Report
A very interesting review by Mehul Patnam et al. mainly focused on the mechanism and therapies targeting corneal lymphangiogenesis. They have discussed some important aspects of previous research and indicated some drawbacks of previous studies. However, there are still some issues which should be addressed before it could be considered for publication.
1. The authors described some potential guidance molecules that play a role in lymphangiogenesis. It would be great if these molecules and their interactions can be illustrated in a schematic to help the readers better understand the contexts.
2. There are significant works related to pathologies involving lymphangiogenesis like alkali burn and infectious keratitis, etc. They are missing from the review.
3. I think that the review's figures are excellent; however, it is not easy to check whether the citation is suitable or not by the present documents because some crucial keywords are not used in the main documents. This point should be improved for readers who may cite this review in the future.
4. The authors focused on the basic principles of lymphatic networks development in mammals in the first half of the article. I think those sections are too long as space is limited. I recommend to involve the basic principles of corneal (lymph)angiogenic privilege. This point should help the readers better understand the breakdown of privilege resulting in pathological lymphatic vessels formation in cornea.
Author Response
"A very interesting review by Mehul Patnam et al. mainly focused on the mechanism and therapies targeting corneal lymphangiogenesis. They have discussed some important aspects of previous research and indicated some drawbacks of previous studies. However, there are still some issues which should be addressed before it could be considered for publication."
1. "The authors described some potential guidance molecules that play a role in lymphangiogenesis. It would be great if these molecules and their interactions can be illustrated in a schematic to help the readers better understand the contexts."
Response: We appreciate the reviewer’s suggestion of including a schematic containing the guidance molecules mentioned in section 3. We have included a summary figure following Table 1 in section 3.
2. "There are significant works related to pathologies involving lymphangiogenesis like alkali burn and infectious keratitis, etc. They are missing from the review."
Response: We agree with the reviewer’s comment that the role of lymphangiogenesis in the pathologies of alkali burn and infectious keratitis is significant and have included additional sections discussing these diseases individually in the section titled “Pathologies Involving Lymphatic Endothelial Guidance.”
3. "I think that the review’s figures are excellent; however, it is not easy to check whether the citation is suitable or not by the present documents because some crucial keywords are not used in the main documents. This point should be improved for readers who may cite this review in the future."
Response: We agree with the reviewer’s concern and have incorporated all crucial keywords in the figure captions as well as throughout the text to maintain consistency.
4. "The authors focused on the basic principles of lymphatic networks development in mammals in the first half of the article. I think those sections are too long as space is limited. I recommend to involve the basic principles of corneal (lymph)angiogenic privilege. This point should help the readers better understand the breakdown of privilege resulting in pathological lymphatic vessels formation in cornea."
Response: We agree with the reviewer’s concern that the discussion of the origin and basic principles of lymphatic networks are too long as space is limited. We have now revised to focus on a more thorough explanation of principles in corneal (lymph)angiogenic privilege.
Reviewer 3 Report
1. The authors have done a lot of work summarizing the lymphangiogenic guidance cues, the underlying mechanisms, their roles in corneal pathologies, and the potential anti-lymphangiogenic therapeutics. But in its current form, the manuscript is not suitable for publication.
2. There is no page number or line number, causing the difficulty of comments.
3. The first word of ABSTRACT and KEYWORDS should not be bold.
4. Incorrect citation format (page 5, line 2; page 7, table 1; page 10, line 24; page 12, line 39; page 21, line 20; page 24, line 28; page 30, line 23)
5. The grammatical tense of some sentences needs to be carefully examined.
6. Page 2, Paragraph 2: The latter two sentences are not related to the beginning of this paragraph, and it is not reasonable to only put the images of the “alkali injury model” here in the introduction, what about other models? You’d better consider putting it in other parts or removing it.
7. Page 13, line 16: “Evidence suggests that the role of Slit/Robo signaling in lymphangiogenesis mirrors its role in axon and vascular endothelial guidance.” Where is the evidence? How to understand that lymphangiogenesis mirrors its role in axon and vascular endothelial guidance? It’s not explained clearly and is hard to understand it.
8. Page 17, lines 3-6: “While Ang-2 is primarily expressed in the human corneal epithelium in normal states, this expression extends to the corneal stroma and endothelium upon inflammation [179]. This dynamic regulation of Ang-2 expression in normal and vascularized corneas implicates Ang-2 as an inducer of corneal lymphangiogenesis.” This expression seems to explain neovascularization rather than lymphangiogenesis. The literature cited here can’t come to the conclusion that Ang-2 is an inducer of corneal lymphangiogenesis.
9. Page 18, paragraph 4: “Notch1 mRNA and protein expression subsided upon administration of an anti-VEGF-A antibody, and a reduction in corneal neovascularization occurred. Furthermore, administration of γ-secretase inhibitor actually promoted VEGF-A expression, despite inhibition of Notch1 pathway activity. These results implicate the Notch signaling pathway in modulating VEGF expression via a negative feedback mechanism: inhibition of Notch signaling and subsequent downregulation of Notch1/Dll4 levels increases VEGF-A expression via feedback signaling.” The latter three sentences are not related to the potential role of Notch1 in lymphangiogenesis. It’s better to remove them.
10. Table 1: Dimethyl fumarate (DMF) 's mechanism of action is unclear. According to the results of this reference, is it activation or inhibition in macrophages? Activation should be inhibition.
11. Page 27, line 11: “The few clinical trials involving corneal lymphangiogenesis have investigated reconstruction of already existing lymphatic malformations without considering lymphatic guidance.” What are the few clinical trials mentioned here?
12. Table 3: This table should be clearer for corneal diseases as this is a review of the cornea which should be underscored.
13. As there are 4 published figures in this manuscript, please be sure to avoid any copyright issues. They should be removed from the current manuscript if it raises potential copyright infringement issues.

Author Response
1. "The authors have done a lot of work summarizing the lymphangiogenic guidance cues, the underlying mechanisms, their roles in corneal pathologies, and the potential anti lymphangiogenic therapeutics. But in its current form, the manuscript is not suitable for publication."
Response: We agree with the reviewer’s concerns and have addressed their comments in the revised manuscript.
2. "There is no page number or line number, causing the difficulty of comments."
Response We have added page numbers and line numbers to each page of the manuscript. Thank you.
3. "The first word of ABSTRACT and KEYWORDS should not be bold."
Response: We have removed the bold type characters from the first word of ABSTRACT and KEYWORDS in the MS Word file.
4. "Incorrect citation format (page 5, line 2; page 7, table 1; page 10, line 24; page 12, line 39; page 21, line 20; page 24, line 28; page 30, line 23)"
Response: We have reviewed all of our citations and have revised the incorrect citation format accordingly.
5. "The grammatical tense of some sentences needs to be carefully examined."
Response: We have carefully examined the grammatical tense of all sentences within the manuscript and these errors are rectified.
6. "Page 2, Paragraph 2: The latter two sentences are not related to the beginning of this paragraph, and it is not reasonable to only put the images of the “alkali injury model” here in the introduction, what about other models? You’d better consider putting it in other parts or removing it."
Response: We agree with the reviewer’s concern and have removed the latter two sentences of Page 2, Paragraph 2 and the image of the “alkali injury model.”
7. "Page 13, line 16: “Evidence suggests that the role of Slit/Robo signaling in lymphangiogenesis mirrors its role in axon and vascular endothelial guidance.” Where is the evidence? How to understand that lymphangiogenesis mirrors its role in axon and vascular endothelial guidance? It’s not explained clearly and is hard to understand it."
Response: We have reviewed the literature and agree with the reviewer’s concern. We have deleted this sentence to ensure this section focuses on Slit/Robo signaling in lymphangiogenesis.
8. "Page 17, lines 3-6: “While Ang-2 is primarily expressed in the human corneal epithelium in normal states, this expression extends to the corneal stroma and endothelium upon inflammation [179]. This dynamic regulation of Ang-2 expression in normal and vascularized corneas implicates Ang-2 as an inducer of corneal lymphangiogenesis.” This expression seems to explain neovascularization rather than lymphangiogenesis. The literature cited here can’t come to the conclusion that Ang-2 is an inducer of corneal lymphangiogenesis."
Response: We agree with the reviewer’s concern that the literature cited here focusing on corneal neovascularization is not sufficient to conclude that Ang-2 is an inducer of corneal lymphangiogenesis. We have deleted these sentences and removed this citation from the section.
9. "Page 18, paragraph 4: “Notch1 mRNA and protein expression subsided upon administration of an anti-VEGF-A antibody, and a reduction in corneal neovascularization occurred. Furthermore, administration of γ-secretase inhibitor actually promoted VEGF-A expression, despite inhibition of Notch1 pathway activity. These results implicate the Notch signaling pathway in modulating VEGF expression via a negative feedback mechanism: inhibition of Notch signaling and subsequent downregulation of Notch1/Dll4 levels increases VEGF-A expression via feedback signaling.” The latter three sentences are not related to the potential role of Notch1 in lymphangiogenesis. It’s better to remove them."
Response: We agree with the reviewer’s concern and have removed the latter three sentences of Page 18, paragraph 4 accordingly.
10. "Table 1: Dimethyl fumarate (DMF) 's mechanism of action is unclear. According to the results of this reference, is it activation or inhibition in macrophages? Activation should be inhibition."
Response: We agree with this point of clarity and have changed the data table to highlight the inhibition of macrophages.
11. "Page 27, line 11: “The few clinical trials involving corneal lymphangiogenesis have investigated reconstruction of already existing lymphatic malformations without considering lymphatic guidance.” What are the few clinical trials mentioned here?"
Response: We have reviewed the literature and found that no clinical trials focusing on corneal lymphatic vessels or corneal lymphangiogenic guidance as an outcome of drug treatment have been completed to date. We are grateful for this clarification and have replaced this statement to accurately reflect this dearth of literature.
12. "Table 3: This table should be clearer for corneal diseases as this is a review of the cornea which should be underscored."
Response: We agree with the reviewer’s concern for clarity and emphasis of corneal diseases in Table 3. We have prioritized the inclusion of trials focusing on corneal diseases and removed unrelated diseases.
13."As there are 4 published figures in this manuscript, please be sure to avoid any copyright issues. They should be removed from the current manuscript if it raises potential copyright infringement issues."
Response: We have ensured that all reproduced figures in the manuscript are free and open access for reproduction in manuscripts.
Reviewer 4 Report
Dears Authors:
The manuscript by Patnam et al. deals with the important topic of lymphatic vessel formation in the cornea. The manuscript provides a profound insight on the mechanisms of lymphangiogenesis in the cornea, including key proteins and signaling pathways involved. I enjoyed reading this very detailed manuscript.
I have the following minor comments:
1.) Page numbers should be included in the manuscript.
2.) Chapter 4.3., Lymphatic vessel malformations: Which implications does the content of this chapter have on lymphangiogenesis in the cornea/corneal diseases or the research in this field? Please provide a link.
Author Response
1. "Page numbers should be included in the manuscript."
Response: We have added page numbers to each page of the manuscript. Thank you.
2. "Chapter 4.3., Lymphatic vessel malformations: Which implications does the content of this chapter have on lymphangiogenesis in the cornea/corneal diseases or the research in this field? Please provide a link."
Response: We agree with the reviewer’s concern regarding Chapter 4.3’s focus on the genetics of lymphatic vessel malformations. We have removed this extraneous section.